# Endometrial Inflammation and Impaired Spontaneous Decidualization: Insights into the Pathogenesis of Adenomyosis

**DOI:** 10.3390/ijerph20043762

**Published:** 2023-02-20

**Authors:** Hiroshi Kobayashi

**Affiliations:** 1Department of Gynecology and Reproductive Medicine, Ms.Clinic MayOne, 871-1 Shijo-cho, Kashihara 634-0813, Japan; hirokoba@naramed-u.ac.jp; Tel.: +81-744-20-0028; 2Department of Obstetrics and Gynecology, Nara Medical University, 840 Shijo-cho, Kashihara 634-8522, Japan

**Keywords:** adenomyosis, decidualization, inflammation, menstruation, microbiota

## Abstract

Adenomyosis is an estrogen-dependent gynecologic disease characterized by the myometrial invasion of the endometrial tissue. This review summarized the current understanding and recent findings on the pathophysiology of adenomyosis, focusing on repeated menstruation, persistent inflammation, and impaired spontaneous decidualization. A literature search was performed in the PubMed and Google Scholar databases from inception to 30 April 2022. Thirty-one full-text articles met the eligibility criteria. Repeated episodes of physiological events (i.e., endometrial shedding, damage, proliferation, differentiation, repair, and regeneration) during the menstrual cycle are associated with inflammation, angiogenesis, and immune processes. The decidualization process in humans is driven by the rise in progesterone levels, independently of pregnancy (i.e., spontaneous decidualization). Adenomyotic cells produce angiogenic and fibrogenic factors with the downregulation of decidualization-associated molecules. This decidualization dysfunction and persistent inflammation are closely related to the pathogenesis of adenomyosis. Recently, it has been found that the reproductive tract microbiota composition and function in women with adenomyosis differ from those without. An increase in opportunistic pathogens and a decrease in beneficial commensals may promote impaired defense mechanisms against inflammation and predispose women to uncontrolled endometrial inflammation. However, currently, there is no direct evidence that adenomyosis is linked to pre-existing inflammation and impaired spontaneous decidualization. Overall, persistent inflammation, impaired spontaneous decidualization, and microbiota dysbiosis (i.e., an imbalance in the composition and function of endometrial microbiota) may be involved in the pathophysiology of adenomyosis.

## 1. Introduction

Adenomyosis is an estrogen-dependent gynecologic disease characterized by the presence of endometrial tissue in the myometrium [1]. Around two-thirds of patients with adenomyosis show abnormal uterine bleeding, pelvic pains, or infertility. Recent advances in molecular biology and animal models have enriched our understanding of the pathogenesis of adenomyosis [2]. Increased estrogen biosynthesis and progesterone resistance have been reported to exert many regulatory functions, including endometrial quality and quantity control [3]. Progesterone resistance is associated with loss of normal endometrial functions (e.g., hemostasis and successful pregnancy), leading to heavy menstrual bleeding, impaired decidualization, nonreceptive endometrial state, and disease establishment and progression [3,4]. Adenomyosis is a multistep disease caused by the accumulation of epigenetic alterations and genetic mutations, dysfunction of decidualization, the promotion of tissue injury caused at the endometrial–myometrial interface, altered expression of angiogenesis-related molecules, the acquisition of invasive traits, and the progression of pathological tissue remodeling in the myometrium [5]. A recent study indicated that the expression of estrogen downstream effectors associated with persistent inflammation, fragile and more permeable vessel formation, and tissue injury and remodeling might be clinically correlated with dysmenorrhea, heavy menstrual bleeding, and infertility, respectively [5]. This review focuses on impaired spontaneous decidualization as a possible mechanism underlying adenomyosis development and discusses why repeated menstruation, which occurs in many modern women, only leads to the development of adenomyosis in a minority of women.

The author first summarizes the molecular mechanisms of fibrotic scar formation after endometrial invagination, a hallmark of adenomyosis, and discusses whether aberrant expression of fibrogenesis-associated molecules has close links with an impaired decidualization process. History of prior uterine surgery (e.g., cesarean delivery, myomectomy, endometrial ablation, dilation and evacuation, and dilation and curettage) has been reported to be an independent risk factor for adenomyosis [6,7]. These data suggest that the invasion of normal endometrial tissue into the myometrium, an artifact of prior uterine manipulation, may contribute to the development of adenomyosis. Moreover, in women without previous uterine surgery, the TIAR (tissue damage and repair) hypothesis and the new EMID (endometrial–fascial interface disruption) hypothesis have been proposed to explain the process of endometrial invagination [2,8]. The concept of the TIAR mechanism was put forward by Leyendecker et al., showing that inner myometrial injury is a result of repetitive microtrauma induced by estrogen, leading to endometrial invagination and adenomyosis development [9]. Frequent and cyclical menstruation and uterine hypercontractility in non-conceptional cycles are crucial in the pathophysiology of endometriosis and adenomyosis [8,10]. Adenomyosis lesions eventually form a fibrotic scar due to repeated injury and repair of the myometrium [11].

Second, this review summarizes the mechanisms by which altered expression of angiogenesis-related molecules impairs proper decidualization. Menstruation is characterized by endometrial tissue destruction, shedding, and regeneration [12]. An essential function of the endometrium is to promote successful implantation and conception. Adequate placentation and embryo implantation for conception require endometrium decidualization [13]. Robust angiogenesis, neovascularization, and vasoconstriction of the spiral arterioles, progressive differentiation of arterioles, an influx and accumulation of immune cells into the endometrium, and accelerating re-epithelialization of the endometrium up to the premenstrual state are essential for cyclic regeneration of the endometrium and decidualization [12].

Then, the author summarizes the evolution of spontaneous decidualization as human conception and its impairment in adenomyosis. The eutherian mammals evolved placentation and intrauterine development of the fetus to introduce and sustain brain development [14]. The decidualization process is essential for embryo implantation. Decidualization in mice is induced by blastocyst attachment to the surface epithelium (i.e., blastocyst-dependent decidualization) [15]. Decidualization caused by maternal signals is a characteristic of non-menstruating species [16,17].

In contrast, in menstruating species such as Homo sapiens, decidualization occurs following the rise in progesterone, independently of pregnancy, and is referred to as spontaneous decidualization [17]. Spontaneous decidualization via progesterone production without the presence of the fetus can improve reproductive efficiency [16]. Progesterone governs decidualization in both non-conception and conception cycles [18]. The differentiation of endometrial stromal cells characterizes human decidualization into decidual cells, maternal spiral artery angiogenesis and remodeling, and an influx of tissue-resident NK cells at the maternal–fetal interface [16,19,20]. Deep trophoblast invasion into maternal uterine blood vessels, where maternal tissues are in direct contact with the semi-allogeneic fetal and paternal antigens, is the hallmark of hemochorial placentation in humans [16]. The immune system is critical in regulating the balance between tolerance (potential survival and growth benefit of the fetus) and defense (a protective mechanism for the mother) against trophoblast invasion [21,22]. Here I focus on impaired spontaneous decidualization associated with disease establishment and progression.

Fourth, the review discusses whether persistent inflammation due to frequent menstrual cycles causes impaired spontaneous decidualization. Primitive menstruating animals experience infrequent menstruation due to repeated reproduction (e.g., pregnancy and lactation) [23]. A woman in hunter-gatherer societies experienced menstruation only about 50 times in her lifetime [24]. Modern women marry later, have fewer children, and breastfeed for shorter periods, resulting in more frequent menstrual cycles [16]. Menstruation may occur as many as 400 times in a lifetime [25]. Therefore, women today menstruate about eight times more often than those in hunter-gatherer societies [26]. Menstruation occurs due to spontaneous decidualization of the endometrium if a woman does not become pregnant [16]. Repeated physiological events (i.e., endometrial cell proliferation, differentiation, shedding, damage, repair, and regeneration) occur across the menstrual cycle, all of which are associated with coordinating inflammatory, immune, and angiogenic processes [26,27,28,29]. Acute tissue injury during menstruation triggers an inflammatory and subsequent healing response that induces transient cytokine and chemokine production. Progesterone withdrawal every month increases the endometrial expression of inflammatory mediators and activates a series of complex signaling cascades of inflammatory pathways due to tissue destruction and further repair.

Finally, this review focuses on defense mechanisms against inflammation and local immune tolerance in the endometrium, as well as their breakdown. In addition to promoting successful implantation and conception, the endometrium’s essential function is protecting the host from infection. Studies reported that microbes often infect the female reproductive tract (i.e., the vagina, cervical canal, uterus, and fallopian tubes) and peritoneum [30]. The inner endometrium of the uterus encounters various symbiotic, commensal biota and pathobiont bacteria [21,30]. Lactobacillus species dominate the vagina [21], but the female reproductive tract and peritoneal cavity harbor more diverse, abundant, and distinct microbial communities [21,31]. The endometrium provides the host defense response to various bacterial pathogens while maintaining the symbiotic relationship with these commensal microbiomes [21]. The author discusses the impact of the commensal microbiome on adenomyosis development.

This review is divided into five parts: fibrogenesis, angiogenesis, decidualization, repeated menstruation, and persistent inflammation, according to the possible pathophysiology of adenomyosis.

## 2. Materials and Methods

Search Strategy and Selection Criteria

A non-systematic literature search was performed to identify relevant studies in PubMed and Google Scholar published from inception to 30 June 2022, using the following keywords: adenomyosis, inflammation, immunity, menstruation, decidualization, angiogenesis, and fibrosis. In the search strategy, these keywords were combined with the Boolean operators AND and OR, as described in Table 1. Inclusion criteria included the publication of original studies and reference lists in review articles. The exclusion criteria were duplicated studies, non-English publications, letters to the editor, poster presentations, and literature unrelated to the research topic. After searching, publications that contained the keyword “endometriosis” were excluded. The identified articles were assessed for eligibility and, subsequently, full-text article assessment. The first identification phase included records identified through a database search (Figure 1). Identified titles and abstracts were screened in the first stage. Duplicates were removed during the second screening phase. Titles, abstracts, and full-text articles were read to remove inappropriate papers. Citation tracking was manually conducted to identify additional relevant articles. The final eligibility phase included the full-text articles for analysis after excluding those for which detailed data could not be extracted.

## 3. Results

### 3.1. Selection of Studies

A literature search was conducted by one researcher (HK), and other investigators did not further review the selected articles for their reliability. The search in the PubMed and Google Scholar electronic databases provided 159 literature citations (Figure 1). Following removing inappropriate articles and duplicates, a total of 73 potentially relevant records were obtained through screening titles and abstracts. Of these, 42 were excluded, and 31 full-text articles met the eligibility criteria.

### 3.2. Fibrotic Lesion Formation Due to Repeated Injury and Repair of the Myometrium

Adenomyosis is characterized by epithelial–mesenchymal transition (EMT), fibroblast-to-myofibroblast trans-differentiation (FMT), and smooth muscle metaplasia (SMM, i.e., the aggregated smooth muscle element within the fibromuscular tissue) [11]. Several studies investigated the gene or protein (Western blot or immunohistochemical) expression of the target molecules in the ectopic and eutopic endometria of women with adenomyosis and in the endometria of patients without adenomyosis, including case-ectopic and case-eutopic tissue samples; case-ectopic and normal endometrium samples; or case-eutopic and normal endometrium samples (Figure 2 and Table 2).Fibrogenesis has been assessed through immunostaining with EMT markers (e.g., E-cadherin, Vimentin, TGF-β, Snail1, Slug, or Snail3), in vitro experiments, and mouse models. Indeed, the expression level of E-cadherin (i.e., a marker for epithelial cells) was decreased in adenomyotic epithelial cells. In contrast, those of vimentin, n-cadherin, and S100A4 (i.e., markers for stromal and mesenchymal cells), proliferating cell nuclear antigen (PCNA), VEGF, and CD31 (i.e., markers of proliferation, angiogenesis, and vasculogenesis) were increased [11,32]. The distribution of collagen deposition in the ectopic endometrial stromata of adenomyosis patients was more extensive compared with their eutopic counterparts and controls [33]. Several key fibrogenesis-related molecules have been identified in adenomyosis lesions, including TGF-β1, phosphorylated Smad3, oxytocin receptor (OXTR), and notch receptor 1 (Notch1), along with increased expression of inflammatory mediators (e.g., IL-1β, IL-6/JAK2/STAT3, corticotropin-releasing hormone [CRH]) and neurogenic (e.g., nerve growth factor [NGF]) [32,34,35].

Molecules related to “Inflammation”, “Spontaneous Decidualization”, “Angiogenesis”, “Fibrosis”, “Energy Homeostasis and Epigenetic Modifications”, and “Neurogenic mediators” are shown in yellow, orange, blue, gray, and light blue, respectively. Target molecules elevated in at least one of the three groups (ectopic versus eutopic, ectopic versus control, and eutopic versus control) were added to the figure [5].

The abbreviations used are as follows: ANXA2, Annexin A2; BDNF, Brain-derived neurotrophic factor; CDK2, Cyclin-dependent kinase 2; CEBPB, CCAAT enhancer binding protein beta, also known as C/EBPβ; COX2, Cyclooxygenase-2; CXCL1, C-X-C motif chemokine ligand 1; Cyclin E; E-cadherin; E2, Estradiol; ERK/MAPK, extracellular signal-regulated kinase/mitogen-activated protein kinase; FOXO1, Forkhead box O1; GM-CSF, Granulocyte-macrophage colony-stimulating factor, also known as colony stimulating factor 2; GRIM-19, Retinoid-interferon (IFN)-induced mortality 19; HIF-1α, Hypoxia-inducible factor-1alpha; HOXA10, HomeoboxA10; HOXA11, HomeoboxA11; IDH1, Isocitrate dehydrogenase 1; IGFBP, Insulin-like growth factor binding protein 1; IL-1R, Interleukin-1R; IL-6, Interleukin-6; IL-8, Interleukin-8; IL-10, Interleukin-10; IL-22, Interleukin-22; JAK, Janus kinase; KISS-1, KiSS-1 metastasis suppressor; KLF5, Kruppel-like factor 5; Lactate, Lactic acid; LIF, Leukemia inhibitory factor; MMP-2, Matrix metalloproteinase-2; MMP-9, Matrix metalloproteinase-9; NF-κB, Nuclear factor kappa B; NGF, Nerve growth factor; NK cells, Natural killer cells; NOTCH1, Notch receptor 1; PGs, Prostaglandins; PRL, Prolactin; RANTES, Regulated upon activation normal T-cell expressed and secreted; SLIT, Slit guidance ligand 2; ROBO, Roundabout 1; SMAD3, Small mother against decapentaplegic (SMAD) family member 3; STAT3, Signal transducer and activator of transcription 3; STIP1, Stress-induced phosphoprotein 1; TF, Tissue factor; TGF-β, Transforming growth factor-β; TLR4, Toll-like receptor 4; TNFR, Tumor necrosis factor receptor; TrkB, Neurotrophic receptor tyrosine kinase 2; and VEGF, Vascular endothelial growth factor.

TGF-β1 promotes fibrosis and decidualization of human endometrial stromal cells by inducing the expressions of COX-2, PGE2, and PRL by activating the extracellular signal-regulated kinase- and Smad-dependent signaling pathways [36]. Furthermore, some evidence suggested that aberrant expression of fibrosis-related molecules such as TGF-β1 might be significantly influenced by abnormal accumulation of macrophages and platelets [37]. Indeed, patients with diffuse adenomyosis showed a marked increase in the density of macrophages and NK cells in the endometrial stroma compared with women with mild focal adenomyosis or women without adenomyosis [46]. Inflammatory cytokines and growth factors secreted by macrophages promote EMT and FMT, eventually leading to lesion invasion and fibrosis [37]. A positive correlation was observed between the number of accumulated macrophages and pain associated with adenomyosis [47]. Moreover, compared with controls, platelet aggregation and myofibroblast accumulation were significantly increased in adenomyosis lesions [11]. Platelet-mediated EMT and FMT often cause fibrosis formation [37]. Guo’s group studies proved that platelet aggregation and activation are crucial in adenomyosis development [11,48,49].

Additionally, the TGF-β superfamily has essential functions in reproduction and development, including oocyte maturation, implantation, and decidualization [15]. The topic here is that overexpression of key fibrogenesis markers TGF-β, Notch1, and OXTR aberrantly regulates the decidualization process [15,42,43,44,45]. TGF-β and its downstream molecules are involved in the pathophysiology of fibrogenic and decidualization processes in adenomyosis. Adenomyosis, characterized by the presence of lesional fibrosis, is closely related to the dysfunction of the decidualization process.

### 3.3. An Altered Expression of Angiogenesis-Related Molecules

This section summarizes how abnormal angiogenesis is associated with aberrant expression of genes responsible for decidualization. Many articles have shown the expression profile of angiogenesis-related molecules in the ectopic endometrium, matched eutopic endometrium, and normal endometrium (Figure 2). Many angiogenic and pro-angiogenic markers are increased in the ectopic and eutopic endometrium compared to the control one: vascular endothelial-derived growth factor (VEGF), HIF-1α, HIF-2α, NF-κB, COX-2, matrix metalloproteinases (MMPs), IL-6, IL-22, TGF-β1, tissue factor (TF), CD41, α-smooth muscle actin (α-SMA), endoglin, S100A13, vimentin, parkinsonism associated deglycase (PARK7, also known as DJ-1), phosphorylated mammalian target of rapamycin (p-mTOR), activin A, follistatin, myostatin, slit guidance ligand 2 (SLIT), roundabout 1 (ROBO1), lysophosphatidic acid (LPA), and 1,4-5, phosphor-signal transducer and activator of transcription 3 (pSTAT3) [50]. Several angiogenesis-related genes are also involved in regulating the decidualization process. Table 3 shows key molecules that positively or negatively affect both the angiogenic and decidualization processes in adenomyosis. Among them, at least the HIF1A [51] and SLIT2 [52] genes potentially impair endometrial decidualization and receptivity. As detailed in the next section, overexpression of the HIF gene, which is most important for angiogenesis, has been reported to reduce endometrial receptivity in experimental animal models via suppression of HOXA10 and HOXA11 [51]. As adenomyosis lesions are related to the expression of several genes that promote or suppress decidualization, the degree of decidualization capabilities may vary from woman to woman, depending on their expression profile.

### 3.4. Dysfunction of Spontaneous Decidualization via Aberrant Epigenetic Alterations

The decidualization of endometrial stromal cells is essential for embryo implantation, placental development, and successful pregnancy. A plethora of molecules is vital in the physiological processes of spontaneous decidualization in the human endometrium [5,14,94] (Figure 2). A transcriptional regulator, HomeoboxA10 (HOXA10), is one of the most promising candidates that can induce a decidualization response [14]. HOXA10 shows a dramatic increase during the mid-secretory phase of the menstrual cycle [14]. Moreover, the HOXA10 is highly expressed in endometrial stromal cells and is regulated under the influence of steroid hormones and by signals from the embryo. In addition to HOXA10, the potential candidate genes affecting decidualization are: HOXA11, insulin-like growth factor binding protein 1 (IGFBP1), prolactin (PRL), forkhead box O1 (FOXO1), CCAAT/enhancer binding protein beta (CEBPB), leukemia inhibitory factor, Indian hedgehog (IHH), Kruppel-like factor 5 (KLF5), heart and neural crest derivatives expressed 2 (HAND2), IL-6, IL-10, and suppressor of cytokine signaling 3 (SOCS3) [5]. Figure 3 shows the mechanism by which progesterone withdrawal causes impaired spontaneous decidualization. In non-conception cycles, progesterone withdrawal reduces the expression of transcriptional regulators (e.g., HOX genes) and angiogenic mediators (e.g., VEGF and TF), causing menstrual bleeding [68]. Menstrual bleeding stimulates thrombin generation, downregulating decidual progesterone receptors and further upregulating many biologically active proinflammatory cytokines and chemokines [68]. Enhanced accumulation of immune cells in the endometrium produces various enzymes involved in the degradation of the extracellular matrix proteins (such as MMPs), further mediating tissue breakdown during menstruation [18]. After the shedding of endometrial tissue, immune cells undergo phenotypic reprogramming from proinflammatory to anti-inflammatory states in response to their environment [18]. In contrast, the loss of progesterone upregulates the expression of proinflammatory cytokines, chemokines, and angiogenesis mediators through enhanced NF-κB transcriptional activation [18]. Persistent inflammation and local hypoxic microenvironment induce excess angiogenesis, trigger proliferation and migration of spiral arterial vascular smooth muscle cells, and cause the development of dilated, fragile blood vessels prone to bleeding [68]. Therefore, repeated menstruation is associated with dysfunction of spontaneous decidualization [16].

This figure shows the mechanism by which progesterone withdrawal causes impaired spontaneous decidualization.

The abbreviations used are as follows: MMP, Matrix metalloproteinase; NF-kB, NF-κB, Nuclear factor kappa B; TF, Tissue factor; and VEGF, Vascular endothelial growth factor.

The decidualization process in women with adenomyosis differs significantly from that in healthy women. The expression of candidate biomarkers associated with the decidualization process was significantly downregulated in the eutopic endometria of patients with adenomyosis compared to the normal endometria of control women [51]. The process was often accompanied by an abnormal increase in endometrial cell proliferation [4]. Reduced expression of these candidate markers causes decidualization failure [14]. Additionally, studies with experimental animal models for adenomyosis revealed that the expression of hypoxia-inducible factor-2α (HIF-2α) in the adenomyosis group was higher than in the control group and that HIF-2α antagonists increased the mRNA and protein expression of HOXA10 and HOXA11 [51]. Thus, the downregulation of HOXA10 and HOXA11 expression by HIF-2α overexpression may be involved in the pathogenesis of adenomyosis in mice [51].

Furthermore, a recent study also provided that aberrant epigenetic alteration might be associated with adenomyosis development [95]. The promoter of progesterone receptor isoform B (PRB), a critical upstream gene for various decidualization-related genes, was hypermethylated in adenomyosis [95]. The downregulation of HOXA10 expression has been reported to be caused by DNA hypermethylation in endometriosis and endometrial cancer [14]. In contrast, C/EBPβ, a transcription factor that regulates cell proliferation, differentiation, and metabolism, has been reported to induce the acetylation of lysine 27 on histone 3 (H3K27ac) modification throughout the genome and upregulate the expression of many genes associated with decidualization, such as IGFBP1 and PRL, via DNA hypomethylation [81] (Figure 2). Therefore, epigenetic modifications may be critical in downregulating decidualization-related genes in adenomyosis. A recent study has shown intriguing results suggesting that dysfunction of spontaneous decidualization may be triggered by the downregulation of decidualization-associated genes, possibly via aberrant epigenetic alterations [5,51,95].

### 3.5. Impaired Spontaneous Decidualization Due to Persistent Inflammation

Abnormal accumulations of immune cells (such as macrophages and lymphocytes) are seen within the eutopic endometrium in women with adenomyosis [5]. Adenomyosis lesions can induce an inflammatory response in the ectopic and eutopic endometrium [96]. Specifically, the adenomyotic lesion environment comprises inflammatory mediators, including proinflammatory cytokines (e.g., IL-6, IL-1β, IL-8), anti-inflammatory cytokines (e.g., IL10, IL-22), transforming growth factor-β [TGF-β], and transcription factor (e.g., nuclear factor kappa B [NF-κB]) [5]. Changes in the expression patterns and local production of inflammatory mediators influence endometrial cell invasion, immune cell recruitment and activation, and crosstalk between immune cells and endometriotic cells [5]. Therefore, adenomyosis is characterized by an imbalance in the expression of pro- and anti-inflammatory cytokines and the downregulation of decidualization-related mediators [5]. The inflammatory system not only affects the immune and endocrine systems but also exhibits crucial interactions with decidualization [97]. The key molecules associated with inflammation (i.e., IL-6, IL-8, NF-κB, IL-22, cyclooxygenase-2 [COX-2], and regulated upon activation, normal T-cell expressed and secreted [RANTES]) and their functions in the decidualization process are summarized in Table 4. Persistent inflammation may represent a cause of spontaneous decidualization dysfunction [26]. Therefore, adenomyosis may be caused by lengthened healthy longevity and excessive lifetime exposure to frequent menstruation and cyclical ovarian hormones. Thus, persistent inflammation caused by repeated menstruation may lead to impaired spontaneous decidualization.

### 3.6. Defense Processes against Inflammation and External Stimuli in the Endometrium

The endometrial barriers of the human female reproductive tract are the first-line defense against invading pathogens and maintain host-microbiota homeostasis. Innate immune cells and endometrial cells recognize invading pathogens or microbes by identifying pathogen-associated molecular patterns (PAMPs; e.g., bacterial cell wall components such as endotoxin or lipopolysaccharide) via host pattern recognition receptors (PRRs; e.g., toll-like receptors [TLRs] and Nod-like receptors) [30]. Macrophages and NK cells initiate, progress, and resolve inflammation via the endometrial TLRs [27]. TLR4 in endometrial cells triggers the production of proinflammatory cytokines and chemokines such as interleukin-6 (IL-6), IL-8, and prostaglandin E2 [30]. The endometrial immune system is an integral part of host defense (e.g., protection against pathogenic organisms); moreover, it also plays a role in the development of immune tolerance to commensal organisms or paternal/fetal antigens (e.g., a successful pregnancy) [21].

Significant differences in the microbiome profile of the vaginal and uterine mucosa between adenomyosis and non-adenomyosis groups have recently been reported [21,31,113]. The endometrium in women with adenomyosis has an increased abundance of opportunistic pathogens and a reduced abundance of beneficial bacteria [31]. The immune response to bacterial infections with opportunistic pathogens depends on the innate and adaptive immune systems and triggers persistent inflammation by activating cytokines and chemokines [30]. The mRNA expression of IL-6 and IL-8 was the highest in the ectopic endometrium for the adenomyosis group, followed by eutopic and normal endometria [114] (Figure 2). The expression of proinflammatory cytokine in adenomyosis may depend on TLRs, as the IL-6 and IL-8 expression was positively correlated with the expression of TLRs [114]. These findings suggest that the eutopic endometria of patients with adenomyosis are recognized as a low-grade inflammatory state due to overexpression of TLRs and that a different expression spectrum of the upper genital tract microbiota may be implicated in disease development. Indeed, the incidence of chronic endometritis was higher in women with adenomyosis (approximately 60%) compared to control women (10%) [115]. Dysbiosis of endometrial microbiota may promote adenomyosis risk, possibly through an aberrant expression of inflammatory cytokines and chemokines (Figure 3). Emerging evidence in this section suggests that certain bacterial infections may be closely associated with several health conditions, such as adenomyosis. Women prone to adenomyosis may share a common inflammatory background. Still, there is no direct evidence that adenomyosis is linked to pre-existing inflammation by specific symbiotic, commensal biota and pathobiont bacteria. 

This review focuses on the current knowledge and recent findings on the mechanisms leading to the dysfunction of spontaneous decidualization among several proposed mechanisms on the pathophysiology of adenomyosis. Moreover, the author discusses the possibility that dysbiosis of endometrial microbiota predisposes women to adenomyosis. Adenomyotic cells either directly produce angiogenic and fibrogenic factors or cooperate with immune cells to produce these mediators. Some angiogenic mediators are essential in the decidualization and fibrogenic processes. In other words, adenomyosis, characterized by lesioned fibrosis and neovascularization, is closely related to the dysfunction of spontaneous decidualization. These findings support immunologic mechanisms by which adenomyosis may interfere with successful implantation and decidualization [46]. Furthermore, dysbiosis of reproductive tract microbiota increased opportunistic pathogens and decreased beneficial commensals, which may promote impaired defense processes against inflammation. In addition, as many as 400 repetitive menstrual cycles cause incessant ovulation across the life course, resulting in more oxidative stress, and may lead to dysfunction of spontaneous decidualization, possibly due to persistent inflammation from frequent endometrial tissue injury and repair [16]. Therefore, repeated menstruation and persistent inflammation may be “two sides of the same coin.” Overall, the author presents the latest advances in the pathogenesis of adenomyosis and suggests for the first time that impaired spontaneous decidualization in women with dysbiosis of reproductive tract microbiota may increase the risk of developing adenomyosis.

The author first discusses why and how repeated menstruation causes impaired decidualization. More frequent reports of menstruation-related disorders, such as dysmenorrhea, hypermenorrhea, anemia, or migraine headaches, are observed in patients with incessant menstruation than in controls [16,116,117]. Furthermore, regular, cyclical menstruation ovulation and excess estrogen exposure increase endometriosis risks, possibly through inflammation, hypoxia, and oxidative stress [116,117]. A similar situation occurs with adenomyosis. In general, upregulation of inflammation-, angiogenesis-, and fibrosis-related molecules and downregulation of decidualization-related molecules (e.g., PRL, IGFBP1, HOXA10, FOXO1, CEBPB, IHH, bone morphogenetic protein 2 (BMP2), nuclear receptor subfamily 2 group F member 2 (NR2F2), Wnt family member 4 (WNT4), and HAND2) are hallmarks of adenomyosis [5,14,94]. Aberrant exposure to cyclical inflammation, immunomodulation, and sex hormones during the reproductive years, and the resulting progesterone resistance, including dysfunction of spontaneous decidualization, may compromise endometrial health and promote adenomyosis development. Any defects in the decidualization process are associated with human reproductive failure (e.g., infertility, recurrent pregnancy loss, placenta accrete, ectopic pregnancy, miscarriage, other obstetric complications, endometriosis, adenomyosis, polycystic ovarian syndrome, dysmenorrhea, and chronic pelvic pain) [14,28,63,118].

Incessant ovulation causes an increased number of menstrual cycles, resulting in more oxidative stress that induces DNA methylation and histone modifications [119]. Indeed, elevated oxidative stress has been detected during the luteal phase of the menstrual cycle [120]. It has been reported that aberrant methylation and histone deacetylation occur in adenomyosis [1,95]. Moreover, the promoter of PRB is hypermethylated in adenomyosis [95]. Altered progesterone signaling through epigenetic modifications is critical in decidualization [121]. Additionally, epigenetic modifications alter the expression of progesterone receptors and many decidualization-related genes in endometrial stromal cells [119]. Specific epigenetic alterations (e.g., hypermethylation that reduces progesterone receptor expression or aberrant gene expression that directly or indirectly acts on progesterone and its downstream targets) can cause progesterone resistance [122].

Interestingly, like adenomyosis, aberrant DNA methylation is critical in the pathogenesis of endometriosis [1]. However, there is no direct evidence that repeated menstruation in women with adenomyosis causes impaired decidualization via epigenetic alterations. Therefore, epigenetic alterations result from inflammation but are not necessarily a cause of adenomyosis. Repeated menstruation may trigger dysfunction of spontaneous decidualization. However, further research is required to examine whether the increased number of menstrual cycles directly affects the expression of various decidualization-related genes through DNA methylation and histone modifications.

Still, it is unclear why repeated menstruation, which occurs in many modern women of childbearing age, only leads to the development of adenomyosis in a minority of women. The once-thought sterile endometrium has been found to harbor diverse microbiota [21,31,113]. A recent study revealed the different vaginal microbiome profiles between women with and without adenomyosis [113]. Moreover, women with adenomyosis have enriched opportunistic pathogens in the upper reproductive tract, and efforts to identify the adenomyosis-specific microbiota are ongoing [31]. Activation of TLR4 by virulence factors produced by specific endometrial microbiota may stimulate the generation of proinflammatory cytokines, such as IL-1β, IL-6, and IL-8, thereby affecting endometrial cell inflammation, immunoregulation, survival, proliferation, invasion, and angiogenesis [27,30,114].

Uncontrolled endometrial inflammation can predispose women to the development of adenomyosis. Therefore, only a minority of women with pre-existing inflammation in the endometrium are more likely to develop adenomyosis. This assumption has been supported by the observation that significantly increased inflammatory cytokine expression and numbers of innate and adaptive immune cells were found in the eutopic endometrium in adenomyosis compared with the normal endometrium [123]. However, it is still unclear whether women with endometrial infections are at an increased risk of developing adenomyosis later in life. For example, gastric cancer is triggered by the complex interplay between bacteria (i.e., Helicobacter pylori), host, and environmental factors [124]. Helicobacter pylori-induced inflammation contributes to neoplastic transformation via oxidative stress, DNA damage, epigenetic modifications, and genetic alterations [124]. Recently, the oxidative stress induced by retrograde menstruation and changes in the pelvic microbiota was deemed likely to promote endometriosis development [125]. Similarly, a minority of women with pre-existing inflammation in the endometrium may be more likely to develop adenomyosis. However, further research is required to identify the adenomyosis-specific microbiota and assess whether excessive and repeated episodes of menstruation in women with prior endometrium inflammation cause impaired decidualization. A multifactorial concept of adenomyosis pathophysiology is illustrated in Figure 4.

The abbreviation used is as follows: TIAR, Tissue damage and repair.

## 4. Conclusions

This review summarizes the current understanding of the association between alterations in the endometrial microenvironment and increased adenomyosis risk. Moreover, updated findings on several mechanisms underlying the pathogenesis of adenomyosis are discussed. Repeated menstruation, incessant ovulation, excessive oxidative stress, and impaired spontaneous decidualization may be crucial in the pathophysiology of adenomyosis. The abnormal processes of angiogenesis, fibrosis, and decidualization are well recognized in the pathogenesis of adenomyosis. Furthermore, this review discusses dysbiosis of reproductive tract microbiota as a possible mechanism underlying adenomyosis development. The author presents a novel concept that women with pre-existing endometrial inflammation and repeated menstruation-induced impaired spontaneous decidualization may be at increased risk of developing adenomyosis, possibly through persistent inflammation and epigenetic alterations. Identifying specific endometrial microbiota and elucidating the molecular mechanisms underlying impaired spontaneous decidualization can provide further knowledge of adenomyosis pathogenesis. In the future, targeted therapies directed against specific molecular drivers that influence the potential relationship between spontaneous decidualization and genital tract microbiota may represent a personalized nonhormonal medicine to improve the life quality of adenomyosis patients.

## Figures and Tables

**Figure 1 ijerph-20-03762-f001:**
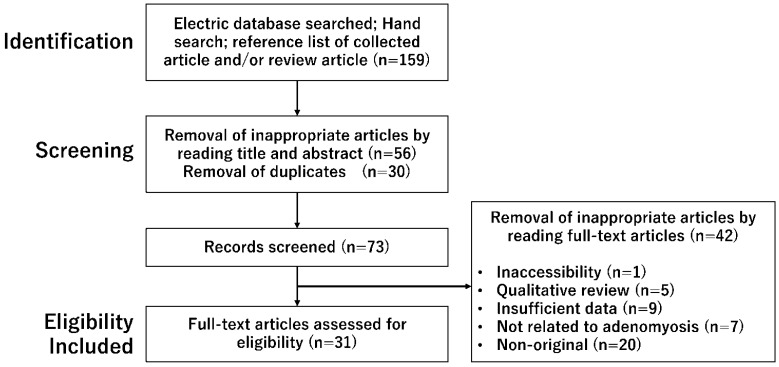
A number of articles were identified by searching using combined keywords. This figure shows the number of published articles identified by combined keywords and the number of records identified through database searching, records after duplicate removal, records screened, removal of inappropriate articles by reading full-text articles, and full-text articles assessed for eligibility.

**Figure 2 ijerph-20-03762-f002:**
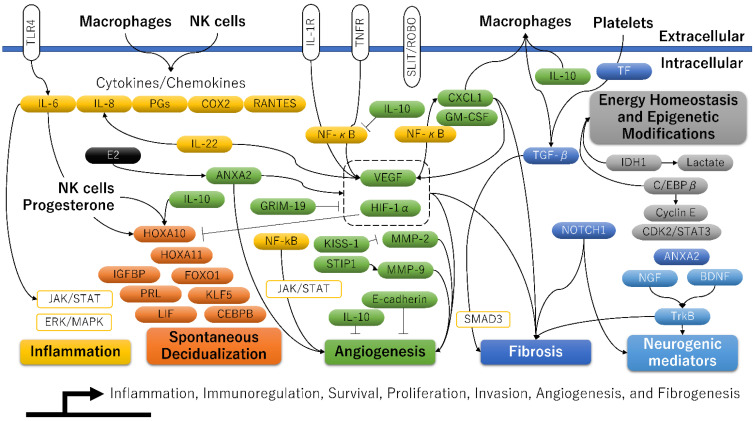
A molecular model related to inflammation, decidualization, angiogenesis, fibrosis, and epigenetic modifications for adenomyosis.

**Figure 3 ijerph-20-03762-f003:**
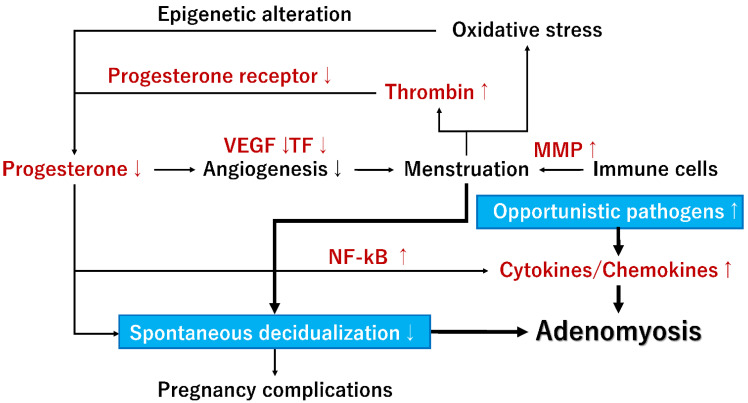
A plausible mechanism by which progesterone withdrawal promotes adenomyosis risk and progression, possibly through the dysfunction of spontaneous decidualization and dysbiosis of endometrial microbiota.

**Figure 4 ijerph-20-03762-f004:**
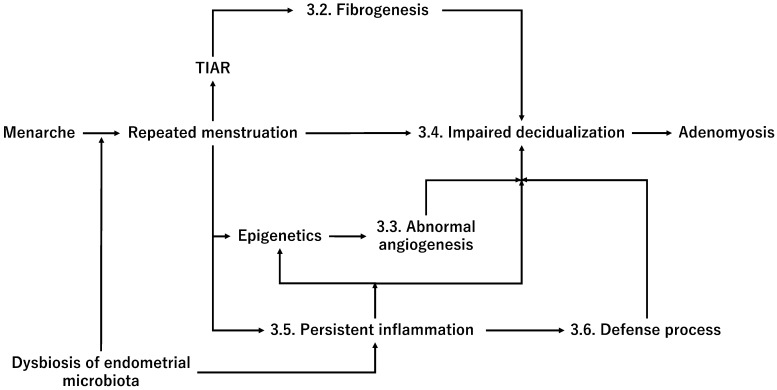
The multifactorial concept of adenomyosis pathogenesis.

**Table 1 ijerph-20-03762-t001:** The search strategy.

Search Mode	The Keyword and Search Term Combinations
Search term 1	Adenomyosis
Search term 2	Inflammation OR Microbiota
Search term 3	Innate immunity OR Adaptive immunity OR Aquired immunity
Search term 4	Menstruation OR Menses
Search term 5	Decidualization OR Decidua OR Implantation
Search term 6	Angiogenesis OR Vasculogenesis
Search term 7	Fibrosis OR Fibrogenesis
Search	Search term 1 AND Search term 2
	Search term 1 AND Search term 3
	Search term 1 AND Search term 4
	Search term 1 AND Search term 5
	Search term 1 AND Search term 6
	Search term 1 AND Search term 7

**Table 2 ijerph-20-03762-t002:** The key molecules associated with fibrogenesis and their functions in adenomyosis. The biological function of each gene is listed in Appendix A.

Official Symbol	Official Full Name	Summary	Refs.
TGFB	transforming growth factor-beta	Smad↑EMT↑FMT↑SMM↑fibrosis↑LIF↑↓; induce or inhibit decidualization	[11,15,33,36,37,38,39,40,41]
OXTR	oxytocin receptor	COX-2↓PGF2α↓; suppression of decidualization process	[42]
NTRK2	neurotrophic receptor tyrosine kinase 2, also known as TrkB	Blastocyst outgrowth↑	[43,44]
NOTCH1	notch receptor 1	Decidualization and implantation↑	[45]

**Table 3 ijerph-20-03762-t003:** The key molecules associated with angiogenesis and their functions in adenomyosis. The biological function of each gene is listed in Appendix A.

Official Symbol	Official Full Name	Summary	Refs.
HIF1A	hypoxia inducible factor 1 subunit alpha	Microvessel density↑fibrogenesis↑VEGF↑HOXA10↓HOXA11↓; suppression of decidualization process	[51,53,54]
VEGFA	vascular endothelial growth factor A	CXCL1↑NF-κB↑M2 polarization↑; promotion of decidualization process	[53,54,55,56,57,58,59,60]
CXCL1	C-X-C motif chemokine ligand 1	Tissue remodeling↑VEGF↑; promotion of decidualization process	[4,61,62]
ANXA2	annexin A2	HIF-1α↑VEGF-A↑β-catenin↑; promotion of decidualization process	[63,64,65,66,67]
F3	coagulation factor III, also known as tissue factor	Thrombin↑PRB↓; stabilization of decidualization due to promotion of hemostasis	[50,68,69,70]
MMP-2, MMP-9	matrix metallopeptidase 2, matrix metallopeptidase 9	Endometrial menstrual breakdown↑	[18,50,56,71,72,73]
IL10	interleukin 10	Anti-angiogenic marker. IL-10↓ during the window of implantation. NF-κB↑HOXA10↓	[50,74]
CDH1	cadherin 1, also known as E-cadherin	Anti-angiogenic marker. MMP2↓MMP9↓; stimulation of the implantation process	[50,75]
SLIT2	slit guidance ligand 2	SLIT-ROBO signaling; suppression of decidualization process	[52,76,77,78,79]
KISS1	KiSS-1 metastasis suppressor	Cell motility in human decidual stromal cells↓; progression of decidualization process	[79,80]
CEBPB	CCAAT enhancer binding protein beta, also known as C/EBPβ	IGFBP-1↑PRL↑; stimulation of decidualization process	[81,82,83]
STIP1	stress-induced phosphoprotein 1	MMP9↑endometrial menstrual breakdown↑	[84]
NDUFA13	NADH:ubiquinone oxidoreductase subunit A13 (NDUFA13), also known as retinoid-interferon (IFN)-induced mortality 19 (GRIM-19)	GRIM19↓(macrophages) IL-1β↑VEGF↑; suppression of decidualization process	[85,86,87,88,89]
CAV1	caveolin 1	CAV1↓IGFBP1↑; stimulation of decidualization process	[90,91]
CSF2	colony stimulating factor 2, also known as GMCSF	Macrophage recruitment	[92,93]

**Table 4 ijerph-20-03762-t004:** The key molecules associated with inflammation and their functions in adenomyosis. The biological function of each gene is listed in Appendix A.

Official Symbol	Official Full Name	Summary	Refs.
IL6, IL8	interleukin 6, interleukin 8	HoxA10↓HoxA11↓MMP↑TIMP↓; suppression of decidualization process	[82,98,99,100,101,102,103]
NFKB1	nuclear factor kappa B subunit 1	LIF↓PRB↓; suppression of decidualization process	[3,64,104,105,106]
IL22	interleukin 22	IL-6↑IL-8↑RANTES↑VEGF↑; suppression of decidualization process	[107,108,109]
PTGS2	prostaglandin-endoperoxide synthase 2, also known as COX-2	IL-6↑IL-8↑; suppression of decidualization process	[110,111]
CCL5	C-C motif chemokine ligand 5, also known as RANTES (regulated upon activation, normal T-cell expressed and secreted)	Trophoblast invasion during early human placentation	[90,112]

## Data Availability

Not applicable.

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
