# Peer review of "Endometrial Inflammation and Impaired Spontaneous Decidualization: Insights into the Pathogenesis of Adenomyosis"

_ijerph, 2023, doi:10.3390/ijerph20043762_

Round 1

Reviewer 1 Report

Comments on IJERPH-2171093

The manuscript is an extensive review of the molecular pathways that have been investigated in adenomyosis and largely reproduces a previous review by the same author (ref 5 of the manuscript), emphasizing the relationship between decidualization and adenomyosis in the present manuscript.

The author claims that impaired decidualization may be involved in the pathogenesis of adenomyosis. However, among the 24 molecules reported as candidates in the pathogenesis of adenomyosis, 4 presented as crucial actors support or stimulate decidualization instead of inhibiting it (ANXA2, CEBPB, CAV1 and PTGS2) while a fifth most important actor, TGF-b1, showed opposite effects. Therefore, conclusions of the review need to be even more cautious than in the present manuscript (for instance, “repeated menstruation, incessant ovulation, … may be crucial in the pathophysiology of adenomyosis” rather than “are crucial” at line 453).

In addition, the number of menstruations is presented as being a factor promoting the occurrence of adenomyosis by preventing decidualization of endometrial stroma. However, the number of menstruations is limited in all women to less than 500 from menarche to menopause. This number cannot increase but can only decrease with multiple pregnancies (or with amenorrheic contraceptive treatment in recent years). But the author does not present any epidemiological data showing that indeed the diagnosis of adenomyosis (confirmed by histopathological analysis of the hysterectomy specimen) is less frequent in multiparous women compared to nulliparous ones. Such data would support a relationship between menstruation (and thus limited decidualization) and adenomyosis.

Minor comments:

-       Line 169: author should explain what is the smooth muscle metaplasia (SMM) occurring in adenomyosis

-       The changes in the expression in the eutopic adenomyotic and/or ectopic endometrium of many molecules presented at Tables 2 and 3 are reported in the text, but not for all molecules. It is therefore difficult to the reader to understand why such molecules are presented as potentially involved in adenomyosis. I would suggest to combine the first two columns of these tables and add columns showing the modification in the expression level of each molecule in the eutopic and ectopic endometrium of adenomyotic uteri.

-       Reference of CEBPB involvement at Table 3 should be 110, not 1100

-       At Fig 3, thrombin production should be increased at menstruation, not decreased

-       A reference other than ref 50 should be added at the end of the sentence at lines 297-300

-       Menstrual cycles do not cause incessant ovulation (line 401). It is rather ovulation that is responsible for menstruation when no fecundation occurs and thus for a menstrual cycle in the human species.

-       Reference 126 for NTRK2 in Supplementary Table 1 is missing

-       MVD should be explained (microvessel density ?) in the summary paragraph related to MMP-2 and MMP-9 in Supplementary Table 2

Author Response

Answer to the reviewers

igerph-2171093

Title: Endometrial inflammation and impaired spontaneous decidualization: insights into the pathogenesis of adenomyosis

Author: Hiroshi Kobayashi

Dear Editor in Chief:

Thank you and the reviewers for the thoughtful comments and helpful suggestions on my manuscript “Endometrial inflammation and impaired spontaneous decidualization: insights into the pathogenesis of adenomyosis” (manuscript ID: igerph-2171093), authored by Hiroshi Kobayashi. I have carefully considered each of the comments, made every effort to address the concerns raised, and applied corresponding revisions to the manuscript. Additionally, I have carefully revised the manuscript to ensure that the text is optimally phrased and free from typographical and grammatical errors. An English proofreading certificate by a native speaker was attached.

The detailed, point-by-point responses to the reviewer comments are given below, whereas the corresponding revisions are highlighted to my manuscript within the document.

I believe that my manuscript has been considerably improved as a result of these revisions, and hope that the revised manuscript is acceptable for publication in Int J Environ Res Public Health.

I would like to thank you once again for your consideration of my work and inviting me to submit the revised manuscript. I look forward to hearing from you.

With best regards,

Hiroshi Kobayashi, M.D., Ph.D.

Department of Gynecology and Reproductive Medicine, Ms.Clinic MayOne, 871-1 Shijo-cho, Kashihara, Nara, 634-0813, Japan

Department of Obstetrics and Gynecology, Nara Medical University, 840 Shijo-cho, Kashihara, Nara, 634-8522, Japan.

Point-by-point responses to reviewer comments

Reviewer 1

Comment 1:

The manuscript is an extensive review of the molecular pathways that have been investigated in adenomyosis and largely reproduces a previous review by the same author (ref 5 of the manuscript), emphasizing the relationship between decidualization and adenomyosis in the present manuscript.

The author claims that impaired decidualization may be involved in the pathogenesis of adenomyosis. However, among the 24 molecules reported as candidates in the pathogenesis of adenomyosis, 4 presented as crucial actors support or stimulate decidualization instead of inhibiting it (ANXA2, CEBPB, CAV1 and PTGS2) while a fifth most important actor, TGF-b1, showed opposite effects. Therefore, conclusions of the review need to be even more cautious than in the present manuscript (for instance, “repeated menstruation, incessant ovulation, … may be crucial in the pathophysiology of adenomyosis” rather than “are crucial” at line 453).

Response 1:

In general, TGFB activates LIF and promotes decidualization, but TGFB can also inhibit the action of LIF. Therefore, the following sentence was modified according to reviewer's indication.

Repeated menstruation, incessant ovulation, excessive oxidative stress, and impaired spontaneous decidualization may be crucial in the pathophysiology of adenomyosis.

Comment 2:

In addition, the number of menstruations is presented as being a factor promoting the occurrence of adenomyosis by preventing decidualization of endometrial stroma. However, the number of menstruations is limited in all women to less than 500 from menarche to menopause. This number cannot increase but can only decrease with multiple pregnancies (or with amenorrheic contraceptive treatment in recent years). But the author does not present any epidemiological data showing that indeed the diagnosis of adenomyosis (confirmed by histopathological analysis of the hysterectomy specimen) is less frequent in multiparous women compared to nulliparous ones. Such data would support a relationship between menstruation (and thus limited decidualization) and adenomyosis.

Response 2:

This review suggests that cyclical bleeding associated with menstruation may be a factor that promotes the development of adenomyosis through impaired spontaneous decidualization of the endometrial stroma. The author could not present any epidemiological data showing whether the histologically confirmed adenomyosis is higher or lower in multiparous women compared to nulliparous ones. The reason is that at least two subtypes of adenomyosis with different localizations have been identified (PMID: 22840719). Classical adenomyosis, also known as type 1 adenomyosis, occurs at the inner myometrium and is characterized by higher ages (multiparous women). Indeed, classical adenomyosis is usually found in multiparous, middle-aged women who present with heavy, painful, cyclical bleeding associated with menstruation (PMID: 7596529), suggesting that the more frequent menstruation, the higher the risk of adenomyosis. In contrast, another type of adenomyosis, also known as type 2 adenomyosis, occurs at the outer myometrium and is more likely to be younger, nulliparous, and have pelvic endometriosis (PMID: 22840719). Therefore, type 1 adenomyosis and type 2 adenomyosis are thought to be more common in multiparous and nulliparous women, respectively. However, this subtype classification is not widely accepted, so this point was not addressed in this manuscript. At present, there are no convincing epidemiologic data to indicate whether the diagnosis of adenomyosis is higher or lower in multiparous women compared with nulliparous women, based on adenomyosis subtyping.

Minor comments:

Comment 3:

-       Line 169: author should explain what is the smooth muscle metaplasia (SMM) occurring in adenomyosis

Response 3:

This sentence was modified according to reviewer's indication.

Adenomyosis is characterized by epithelial-mesenchymal transition (EMT), fibro-blast-to-myofibroblast trans-differentiation (FMT), and smooth muscle metaplasia (SMM, i.e., the aggregated smooth muscle element within the fibromuscular tissue) [11].

Comment 4:

-       The changes in the expression in the eutopic adenomyotic and/or ectopic endometrium of many molecules presented at Tables 2 and 3 are reported in the text, but not for all molecules. It is therefore difficult to the reader to understand why such molecules are presented as potentially involved in adenomyosis. I would suggest to combine the first two columns of these tables and add columns showing the modification in the expression level of each molecule in the eutopic and ectopic endometrium of adenomyotic uteri.

Response 4:

The expression level of each molecule was measured by RT-PCR, Western blot, immunostaining, etc. and varies from literature to literature. The expression levels of each molecule were compared according to the following three patterns; ectopic endometrium versus eutopic endometrium, ectopic endometrium versus control endometrium, and eutopic endometrium versus control endometrium. However, the literature containing all three patterns is limited. For example, below are the expression levels of TGFB (first molecule in Table 2) and HIF1A (first molecule in Table 3). Different publications give different results. Or the required data is not measured. Therefore, it was difficult to compare and display expression levels in ectopic, eutopic, and control endometrium separately. I think this result is more confusing for the reader.

Official Symbol/Official Full Name

Ectopic/Eutopic

Ectopic/Control

Eutopic/Control

Summary

Refs.

Table 2

TGFB, transforming growth factor-beta

↑→→↑↑↑

↑↑↑↑??

↑↑↑→??

Smad↑EMT↑FMT↑SMM↑fibrosis↑LIF↑↓; induce or inhibit decidualization

11,33,15,120,121,122,123

Omitted below

Table 3

HIF1A, hypoxia inducible factor 1 subunit alpha

↑→↑

↑↑?

↑↑?

Microvessel density↑ fibrogenesis↑ VEGF↑ HOXA10↓ HOXA11↓; suppression of decidualization process

45,85,86

Omitted below

I added the following sentence to the subsection 3.2.:

Several studies investigated the gene or protein (Western blot or immunohistochemical) expression of the target molecules in the ectopic and eutopic endometrium of women with adenomyosis and in the endometrium of patients without adenomyosis, including case-ectopic and case-eutopic tissue samples; case-ectopic and normal endometrium samples; or case-eutopic and normal endometrium samples (Figure 2, and Table 2).

Figure 2 legend:

Molecules related to 'Inflammation', 'Spontaneous Decidualization', 'Angiogenesis', 'Fibrosis', 'Energy Homeostasis and Epigenetic Modifications', and 'Neurogenic mediators' are shown in yellow, orange, blue, gray, and light blue, respectively. Target molecules elevated in at least 1 of the 3 groups (ectopic versus eutopic, ectopic versus control, and eutopic versus control) were added to the figure [5].

Comment 5:

-       Reference of CEBPB involvement at Table 3 should be 110, not 1100

Response 5:

Fixed a mistake.

Comment 6:

-       At Fig 3, thrombin production should be increased at menstruation, not decreased

Response 6:

Fixed a mistake.

Comment 7:

-       A reference other than ref 50 should be added at the end of the sentence at lines 297-300

Response 7:

This sentence was modified according to reviewer's indication.

[5,45,49]

Comment 8:

-       Menstrual cycles do not cause incessant ovulation (line 401). It is rather ovulation that is responsible for menstruation when no fecundation occurs and thus for a menstrual cycle in the human species.

Response 8:

This sentence was modified according to reviewer's indication.

Incessant ovulation causes an increased number of menstrual cycles, resulting in more oxidative stress that induces DNA methylation and histone modifications [60].

Comment 9:

-       Reference 126 for NTRK2 in Supplementary Table 1 is missing

Response 9:

I corrected because the citation number was off by one.

Comment 10:

-       MVD should be explained (microvessel density ?) in the summary paragraph related to MMP-2 and MMP-9 in Supplementary Table 2

Response 10:

The word "microvessel density (MVD)" was inserted.

Reviewer 2 Report

Comments to the Authors:

This manuscript summarized the current understanding and recent findings on the pathophysiology of adenomyosis, focusing on repeated menstruation, persistent inflammation, and impaired spontaneous decidualization. Moreover, updated findings on several mechanisms underlying the pathogenesis of adenomyosis are discussed. This is a very meaningful work, which can provide ideas for the current work of researchers in this field. This manuscript can be published in Int. J. Environ. Res. Public Health, however, some revisions are needed before this article can be published.

1.      There is a colloquial phenomenon in the language of the article, such as line 68, page 2 “Second, I summarize the mechanisms by which altered expression of angiogenesis-related molecules impairs proper decidualization”; line 78, page 2 “Then, I summarize the evolution of spontaneous decidualization as human conception and its impairment in adenomyosis”. Carefully check the language in the paper for standardization, including the use of abbreviations, and grammatical issues.

2.      The format of references is not uniform. For example, Ref. 15, change “Ni N” to “Ni N.”; Ref. 58, change “Kim .CJ.” to “Kim C.J.”.

3.      The summary of the research background in the introduction is not deep enough. Please enrich the introduction and briefly summarize the main content of this paper.

4.      Introduction, the author introduces some treatments for adenomyosis of uterus and related diseases. Supramolecular strategies should be mentioned. In support of this view, the following important recent papers should be cited: Chem. Soc. Rev. 2021, 50, 2839; Sci. China: Chem. 2023, DOI: 10.1007/s11426-022-1477-x.

Author Response

Answer to the reviewers

igerph-2171093

Title: Endometrial inflammation and impaired spontaneous decidualization: insights into the pathogenesis of adenomyosis

Author: Hiroshi Kobayashi

Dear Editor in Chief:

Thank you and the reviewers for the thoughtful comments and helpful suggestions on my manuscript “Endometrial inflammation and impaired spontaneous decidualization: insights into the pathogenesis of adenomyosis” (manuscript ID: igerph-2171093), authored by Hiroshi Kobayashi. I have carefully considered each of the comments, made every effort to address the concerns raised, and applied corresponding revisions to the manuscript. Additionally, I have carefully revised the manuscript to ensure that the text is optimally phrased and free from typographical and grammatical errors. An English proofreading certificate by a native speaker was attached.

The detailed, point-by-point responses to the reviewer comments are given below, whereas the corresponding revisions are highlighted to my manuscript within the document.

I believe that my manuscript has been considerably improved as a result of these revisions, and hope that the revised manuscript is acceptable for publication in Int J Environ Res Public Health.

I would like to thank you once again for your consideration of my work and inviting me to submit the revised manuscript. I look forward to hearing from you.

With best regards,

Hiroshi Kobayashi, M.D., Ph.D.

Department of Gynecology and Reproductive Medicine, Ms.Clinic MayOne, 871-1 Shijo-cho, Kashihara, Nara, 634-0813, Japan

Department of Obstetrics and Gynecology, Nara Medical University, 840 Shijo-cho, Kashihara, Nara, 634-8522, Japan.

Point-by-point responses to reviewer comments

Reviewer 2

This manuscript summarized the current understanding and recent findings on the pathophysiology of adenomyosis, focusing on repeated menstruation, persistent inflammation, and impaired spontaneous decidualization. Moreover, updated findings on several mechanisms underlying the pathogenesis of adenomyosis are discussed. This is a very meaningful work, which can provide ideas for the current work of researchers in this field. This manuscript can be published in Int. J. Environ. Res. Public Health, however, some revisions are needed before this article can be published.

Comment 1:

  1. There is a colloquial phenomenon in the language of the article, such as line 68, page 2 “Second, I summarize the mechanisms by which altered expression of angiogenesis-related molecules impairs proper decidualization”; line 78, page 2 “Then, I summarize the evolution of spontaneous decidualization as human conception and its impairment in adenomyosis”. Carefully check the language in the paper for standardization, including the use of abbreviations, and grammatical issues.

Response 1:

These sentences were modified according to reviewer's indication.

Comment 2:

  1. The format of references is not uniform. For example, Ref. 15, change “Ni N” to “Ni N.”; Ref. 58, change “Kim .CJ.” to “Kim C.J.”.

Response 2:

Fixed a mistake.

Comment 3:

  1. The summary of the research background in the introduction is not deep enough. Please enrich the introduction and briefly summarize the main content of this paper.

Response 3:

The following sentence was added at the end of the first paragraph of the introduction section:

This review focuses on impaired spontaneous decidualization as a possible mechanism underlying adenomyosis development and discusses why repeated menstruation, which occurs in many modern women, only leads to the development of adenomyosis in a minority of women.

Furthermore, the following sentence was added at the end of the introduction section:

This review is divided into five parts: fibrogenesis, angiogenesis, decidualization, repeated menstruation, and persistent inflammation, according to the possible pathophysiology of adenomyosis.

The introduction section was revised to further clarify the direction of this review.

Comment 4:

  1. Introduction, the author introduces some treatments for adenomyosis of uterus and related diseases. Supramolecular strategies should be mentioned. In support of this view, the following important recent papers should be cited: Chem. Soc. Rev. 2021, 50, 2839; Sci. China: Chem. 2023, DOI: 10.1007/s11426-022-1477-x.

Response 4:

The rapid development of supramolecular strategies will advance the field of therapeutics. However, we believe that a therapeutic strategy based on a supramolecular strategy for adenomyosis is an issue for the future. Even in the field of endometriosis, where basic research is more advanced than adenomyosis, diagnosis and treatment based on supramolecular strategies is still in its infancy. I think it is premature to discuss it here, so I will refrain from introducing therapeutic strategies based on supramolecular strategies. However, the following sentence was added at the end of the Conclusion section:

In the future, targeted therapies directed against specific molecular drivers that influence the potential relationship between spontaneous decidualization and genital tract microbiota may represent a personalized nonhormonal medicine to improve the life quality of adenomyosis patients.

Reviewer 3 Report

Present study is about the pathogenesis of adenomyosis which is not a life threatening disease. However, this review seems novel due to the advance and diverse sort of methodology presentation. The data and references are fullfilling the requirement of a review article. Presentation of results is also effective to some extent. But how can you discuss the results under the discussion section as there are not any to be compared with literature. So, kindly follow some important recommendation from reviewers side.

Title:

The title is good and comprehensive.

Abstract:

There is need to thoroughly revise the abstract. It is written in in appropriate way. Like firstly you are talking about decidualization but at once, you started about the dysbiosis. Kindly rewrite the abstract after the point of spontaneous decidualization.

The dysbiosis is only mentioned in 3.6 sub-section so describe it accordingly in only 1-2 lines in abstract.

Results:

1.      In Figure 2 and 3 legend, please mention full names of each abbreviation.

2.      Write Figure legends under the Figures.

Discussion:

            The discussion section in a review is inappropriate. As you don’t have any novel or latest finding which can be compared with those of previous literature. Also the discussion is focusing what the author has done in this review which is not playing any role in improving the quality of review.

Kindly remove the heading discussion. And incorporate the whole data in results section. As it is not an original article so there is no strict need of dividing it into methodology, results and discussion sections. However, methodology and results headings can be considered.

Figure 4 should be incorporated in the epidemiology section.

Author Response

Answer to the reviewers

igerph-2171093

Title: Endometrial inflammation and impaired spontaneous decidualization: insights into the pathogenesis of adenomyosis

Author: Hiroshi Kobayashi

Dear Editor in Chief:

Thank you and the reviewers for the thoughtful comments and helpful suggestions on my manuscript “Endometrial inflammation and impaired spontaneous decidualization: insights into the pathogenesis of adenomyosis” (manuscript ID: igerph-2171093), authored by Hiroshi Kobayashi. I have carefully considered each of the comments, made every effort to address the concerns raised, and applied corresponding revisions to the manuscript. Additionally, I have carefully revised the manuscript to ensure that the text is optimally phrased and free from typographical and grammatical errors. An English proofreading certificate by a native speaker was attached.

The detailed, point-by-point responses to the reviewer comments are given below, whereas the corresponding revisions are highlighted to my manuscript within the document.

I believe that my manuscript has been considerably improved as a result of these revisions, and hope that the revised manuscript is acceptable for publication in Int J Environ Res Public Health.

I would like to thank you once again for your consideration of my work and inviting me to submit the revised manuscript. I look forward to hearing from you.

With best regards,

Hiroshi Kobayashi, M.D., Ph.D.

Department of Gynecology and Reproductive Medicine, Ms.Clinic MayOne, 871-1 Shijo-cho, Kashihara, Nara, 634-0813, Japan

Department of Obstetrics and Gynecology, Nara Medical University, 840 Shijo-cho, Kashihara, Nara, 634-8522, Japan.

Point-by-point responses to reviewer comments

Reviewer 3

Present study is about the pathogenesis of adenomyosis which is not a life threatening disease. However, this review seems novel due to the advance and diverse sort of methodology presentation. The data and references are fullfilling the requirement of a review article. Presentation of results is also effective to some extent. But how can you discuss the results under the discussion section as there are not any to be compared with literature. So, kindly follow some important recommendation from reviewers side.

Comment 1:

Title:

The title is good and comprehensive.

Response 1:

Thank you.

Comment 2:

Abstract:

There is need to thoroughly revise the abstract. It is written in in appropriate way. Like firstly you are talking about decidualization but at once, you started about the dysbiosis. Kindly rewrite the abstract after the point of spontaneous decidualization.

Response 2:

In the Abstract section, the sentence "Furthermore, the reproductive tract microbiota composition and function in women with adenomyosis differ from those without" was removed and added the following sentence:

This decidualization dysfunction and persistent inflammation are closely related to the pathogenesis of adenomyosis. Recently, it has been found that the reproductive tract microbiota composition and function in women with adenomyosis differ from those without.

Comment 3:

The dysbiosis is only mentioned in 3.6 sub-section so describe it accordingly in only 1-2 lines in abstract.

Response 3:

Overall, persistent inflammation, impaired spontaneous decidualization, and microbiota dysbiosis (i.e., an imbalance in the composition and function of endometrial microbiota) may be involved in the pathophysiology of adenomyosis.

Comment 4:

Results:

  1. In Figure 2 and 3 legend, please mention full names of each abbreviation.

Response 4:

Figure legends were modified according to reviewer's indication.

Comment 5:

  1. Write Figure legends under the Figures.

Response 5:

Figure legends were modified according to reviewer's indication.

Comment 6:

Discussion:

The discussion section in a review is inappropriate. As you don’t have any novel or latest finding which can be compared with those of previous literature. Also the discussion is focusing what the author has done in this review which is not playing any role in improving the quality of review.

Response 6:

The following sentence was added to the end of the first paragraph of the discussion section as it is a new finding for this review:

Overall, the author presented the latest advances in the pathogenesis of adenomyosis and suggested for the first time that impaired spontaneous decidualization in women with dysbiosis of reproductive tract microbiota may increase the risk of developing adenomyosis.

Comment 7:

Kindly remove the heading discussion. And incorporate the whole data in results section. As it is not an original article so there is no strict need of dividing it into methodology, results and discussion sections. However, methodology and results headings can be considered.

Response 7:

The heading "discussion" was removed.

Comment 8:

Figure 4 should be incorporated in the epidemiology section.

Response 8:

Figure 4 was incorporated in the epidemiology section according to reviewer's indication.
